

# Text data augmentation and pre-trained Language Model for enhancing text classification of low-resource languages

Atabay Ziyaden[1,2], Amir Yelenov[2,3], Fuad Hajiyev[4], Samir Rustamov[4] and Alexandr Pak[1,2]

[1] Kazakh-British Technical University, Almaty, Kazakhstan
[2] Institute of Information and Computational Technologies, Almaty, Kazakhstan
[3] Nazarbayev University, Astana, Kazakhstan
[4] School of Information Technologies and Engineering, ADA University, Baku, Azerbaijan

## ABSTRACT

**Background:** In the domain of natural language processing (NLP), the development and success of advanced language models are predominantly anchored in the richness of available linguistic resources. Languages such as Azerbaijani, which is classified as a low-resource, often face challenges arising from limited labeled datasets, consequently hindering effective model training.

**Methodology:** The primary objective of this study was to enhance the effectiveness and generalization capabilities of news text classification models using text augmentation techniques. In this study, we solve the problem of working with low-resource languages using translations using the Facebook mBart50 model, as well as the Google Translate API and a combination of mBart50 and Google Translate thus expanding the capabilities when working with text.

**Results:** The experimental outcomes reveal a promising uptick in classification performance when models are trained on the augmented dataset compared with their counterparts using the original data. This investigation underscores the immense potential of combined data augmentation strategies to bolster the NLP capabilities of underrepresented languages. As a result of our research, we have published our labeled text classification dataset and pre-trained RoBERTa model for the Azerbaijani language.

# INTRODUCTION

In the realm of natural language processing (NLP), the development of sophisticated language models has revolutionized the way we interact with and understand textual data. Transformer-based (*Vaswani et al., 2023*) architectures, such as BERT (*Devlin et al., 2019*), GPT (*Pouran Ben Veyseh et al., 2021*), and RoBERTa (*Liu et al., 2019*) have demonstrated exceptional capabilities in a multitude of language-related tasks, ranging from sentiment analysis to machine translation. However, the effectiveness of these models is largely dependent on the availability of vast and diverse training datasets. This presents a

Corresponding author
Atabay Ziyaden,
iamdenay@gmail.com

significant challenge when dealing with low-resource languages, which are languages with scant linguistic resources that hinder the effective development of language models.

Azerbaijani, a Turkic language spoken by millions of people, falls under the category of low-resource languages in the NLP landscape. While there exist some resources such as word2vec and GloVe embeddings for Azerbaijani (*Huseynov et al., 2020*), a notable gap remains: the absence of transformer-based language models tailored to the nuances of the language. Despite their prominence and cultural significance, the dearth of pre-trained models impedes progress in various NLP tasks, including text classification.

This study was motivated by the absence of advanced language models for the Azerbaijani language. Our research seeks to bridge this gap by exploring innovative techniques to enhance the performance of Azerbaijani text classification, leveraging the limited available resources. Based on our comprehensive analysis of the literature, it appears that no augmentation was performed in the Azerbaijani language. Through augmentation and fine-tuning strategies, we aimed to develop robust models capable of improving text classification tasks in Azerbaijani.

The primary constraint in tackling this challenge is the lack of labeled data for Azerbaijani text classification. The absence of substantial corpora for training specialized models hampers the effective training of classification algorithms. Augmentation techniques have emerged as promising solutions for mitigating the limitations of data scarcity. Augmentation involves generating synthetic data instances from an existing dataset while preserving the underlying semantic meaning. These techniques have the potential to expand the dataset, enhance the generalization capabilities of the model, and boost classification performance.

Our study addresses this gap by focusing on LRLs, employing a novel integration of the Facebook mBart50 model and the Google Translate API, both independently and in conjunction, to enhance text augmentation in these languages. Unlike previous works, our approach combines the strengths of advanced machine translation and deep learning models to overcome the challenges of data scarcity and linguistic diversity inherent in LRLs. This methodology not only contributes to the field of NLP but also aligns with the broader goal of promoting linguistic inclusivity in digital communication systems. Our study thus provides a distinct perspective in the landscape of text augmentation research, emphasizing the need for and feasibility of advanced solutions in the context of LRLs.

The following is a summary of the study's main contributions:

- Introduced the first publicly available transformer-based language model for Azerbaijani language.
- Proposed the labelled dataset of Azerbaijani News.
- First, to the best of our knowledge, demonstrated the effective application of various text augmentation techniques for enhancement news classification task in Azerbaijani language.
- Observed that the augmentation with Google Translate achieved a notable result with a score of 0.87, indicating a 0.04 improvement.

- Evidenced the effectiveness of Neural Machine Translation (mBart-50) with an F1-score of 0.86, highlighting its competitive quality against Google Translation Service while offering faster processing and cost-efficiency.
- The mixed approach resulted in an accuracy of 0.86, closely aligned with the mBart-50 translation strategy, suggesting limited benefits relative to the resources and time expended.

The next sections of the study describes the methodology employed, including the dataset, augmentation techniques, and evaluation metrics. Subsequently, the results of our experiments are presented, highlighting the impact of augmentation on classification performance. Finally, we engage in a comprehensive discussion of the findings, elucidate their implications, and suggest future research directions.

## RELATED WORK

Prior to the deep learning era, text augmentation employed different methodologies. *Riezler et al. (2007)* leveraged statistical machine translation for query expansion, demonstrating the efficacy of translation-based methods in dataset enhancement. However, these methods often lacked the nuanced understanding of context and language syntax. *Nigam et al. (2000)* presented a method combining labeled and unlabeled documents for classification using the EM algorithm, which, while innovative, faced limitations in handling highly complex or ambiguous data.

The advancement of NLP led to a paradigm shift in text augmentation, with the introduction of various techniques such as synonym replacement, random operations, and back-translation (*Wei & Zou, 2019*; *Abonizio, Paraiso & Barbon, 2022*; *Feng et al., 2022*; *Karimi, Rossi & Prati, 2021*). While these methods have shown effectiveness, their application is often constrained in low-resource languages, where tools akin to WordNet *Miller (1995)* for tasks like synonym or hyponym replacement are scarce or non-existent. This limitation leads to significant challenges in maintaining the linguistic accuracy and cultural relevance of augmented texts in these languages.

SeqGAN (*Luo, Bouazizi & Ohtsuki, 2021*) and Data Boost (*Liu et al., 2020*) explored GAN and reinforcement learning approaches, respectively, but often required extensive computational resources and could produce unpredictable results in terms of data quality.

Applications of augmentation in sentiment analysis (*Abdurrahman & Purwarianti, 2019*; *Tang, Tang & Yuan, 2020*; *Chen et al., 2017*) and extractive summarization (*Bacco et al., 2021*) have shown significant improvements in NLP model performance. However, these techniques often struggle with maintaining the semantic integrity of the original text, which is crucial in such tasks. The innovative approaches of G-DAUGC (*Yang et al., 2020*) and TextFooler (*Jin et al., 2020*) highlighted the transformative potential of data augmentation, yet their applicability in real-world scenarios remains a subject of ongoing research.

The existing literature, while comprehensive in exploring various augmentation techniques, often overlooks the specific challenges associated with low-resource languages (LRLs). This gap is particularly evident in neural data-to-text generation (*Chang et al.,*

*2021*) and context-based augmentation methods (*Sharifirad, Jafarpour & Matwin, 2018*), where the scarcity of data in LRLs limits the effectiveness of these approaches.

Text augmentation has evolved significantly, with varied methods targeting distinct challenges. Collectively these efforts demonstrate the potential of augmentation techniques for refining and enhancing NLP models across diverse applications. In addition, this methods are direct answer to tasks arising from LRLs., mirroring a broader academic and societal interest in fostering inclusivity and diversity in digital communication systems. The issue of LRLs has emerged as a focal point of inquiry and engagement within the scientific community (*Shafiq et al., 2023*; *Karyukin et al., 2023*; *Sazzed, 2021*; *Ramponi et al., 2022*; *Farooq et al., 2023*). Addressing the challenges posed by LRLs is seen as a pivotal step towards achieving linguistic equity in the digital domain.

## DATA

### Dataset composition

Our dataset encompasses a comprehensive collection of Azerbaijani news texts from the *Azertac (2023)* State Agency, drawn from a variety of news articles. Azertac, established on March 1, 1920, was recognized as a pioneering entity within the framework of international information agencies. It has played a pivotal role in the establishment and coordination of various associations, including the Association of National Information Agencies comprising nations affiliated with the Commonwealth of Independent States, the Association of News Agencies representing Turkish-speaking countries, and the Association of National News Agencies associated with countries participating in the Black Sea Economic Cooperation Organization. AZERTAC has engaged in collaborative endeavors with several renowned news agencies to foster global information exchange and cooperation. This extensive network of collaborations underscores Azertac's global reach and influence in international news dissemination.

The dataset comprises approximately three million rows, with each row representing a sentence extracted from diverse Azerbaijani news sources. These sentences cover a wide spectrum of subjects, including but not limited to politics, the economy, culture, sports, technology, and health. The Labeled dataset, which has been posted and publicly shared in the link, is organized to facilitate rigorous analysis and classification tasks, with essential metadata provided for each sentence. In these studies, additional information about the collection and use of this dataset can be found (*Kalbaliyev & Rustamov, 2021*; *Huseynov et al., 2020*; *Suleymanov & Rustamov, 2018*). The dataset is available on our Zenodo (*AzerTac, 2023*).

### Metadata information

The dataset is enriched with crucial metadata attributes that enhance its utility and applicability to various research tasks:

- **News category**: Each sentence is linked to a specific news category, covering subjects such as politics, economy, culture, sports, technology, and health.

- **News subcategory**: Further enhance granularity, each sentence is classified into a subcategory, enabling fine-tuned analysis and specialized classification tasks.
- **News index**: A unique identifier for each news article maintains the dataset integrity and supports cross-referencing.
- **News sentence order**: Sequential order aids in preserving sentence context, which is essential for text generation and summarization.
- **Link**: Hyperlinks to original articles provide direct access for researchers to delve into the sentence context.
- **Sentence**: The core textual content, which varies in length and complexity, covers a spectrum of linguistic styles and themes.

## Class distribution and imbalance analysis

An exploratory data analysis (EDA) revealed a class imbalance in the dataset. The distribution of sentences across news categories was notably skewed, with some categories having significantly more instances than others. For instance, the "CƏMİYYƏT" (Society) category accounts for the highest number of sentences (approximately 1,074,514), while categories like "ŞUŞA İLİ" (Year of Shusha) have substantially fewer instances (only 40 sentences) which could be seen in Fig. 1.

## Word popularity

As part of our dataset analysis, we examined the most frequently occurring non-stop words in the Azerbaijani news texts. These words provide valuable insights into the language characteristics and prevailing themes within the dataset. The selection of the most prominent words, along with their associated frequencies as sizes, is shown in Fig. 2.

These words represent a cross-section of the linguistic and thematic landscape within the Azerbaijani news articles. The associated frequencies indicate the prevalence of these words in the dataset, shedding light on the frequently discussed topics and ideas. For instance, words like 'azərbaycan' and 'dövlət' suggest a focus on national and governmental matters, while terms such as 'siyasi' and 'fəaliyyət' point to political and operational discussions.

Analyzing these popular words and their frequencies can guide our understanding of the core themes of the dataset, which in turn informs the design and effectiveness of our classification models. By leveraging these insights, we aim to create models that better capture the nuances and contexts inherent in Azerbaijani news texts. One of the challenges of the Azerbaijani language is agglutinative language. Each part of speech in this language can be modified by suffixes and prefixes. For example, the word "book" can generate around 400 new grammatical and lexical words which creates huge sparsity in the word space.

## Examples and implications

The class imbalance, which can be seen in Table 1, in the dataset and the varying sentiment distribution across news categories present challenges for the development of effective
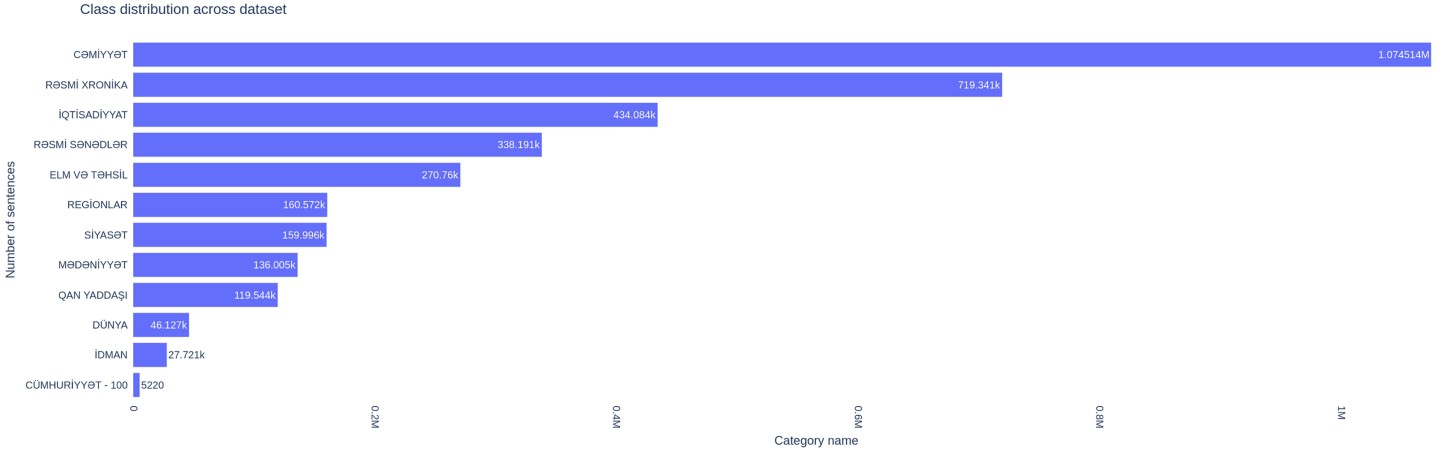

**Figure 1** Distribution of sentences across different classes.

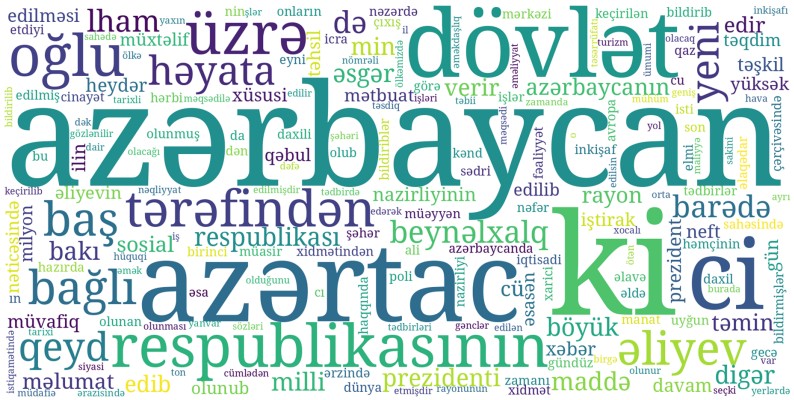

**Figure 2** Word cloud of most popular words.

augmentation and classification models. Addressing these challenges is crucial for ensuring robust performance and unbiased results. Strategies such as oversampling, undersampling, and synthetic data generation can be employed to mitigate the impact of class imbalance. Furthermore, sentiment distribution insights can guide the development of sentiment-aware models that account for the nuances of sentiment representation across different categories.

In the following sections, we detail our methodology for data augmentation, training, and evaluation, with a focus on tackling these challenges to achieve accurate and reliable results.

## METHODS

The primary objective of this study is to enhance the effectiveness and generalization capabilities of news text classification models through text augmentation techniques. Given the absence of readily available language models for Azerbaijani, we performed pre-training on a significant *corpus*.

**Table 1 Class size in relation to largest class.**

| Class name | % of largest class | Description |
|---|---|---|
| Cəmiyyət | 100 | Societal issues, human interest stories |
| Rəsmi xronika | 66.946 | Official events, governmental announcements |
| İqtisadiyyat | 40.398 | Economic matters, business activities |
| Rəsmi sənədlər | 31.474 | Official documents and administrative reports |
| Elm və Təhsil | 25.198 | Educational activities and advancements in the field of science |
| Regionlar | 14.944 | Regional news, local events |
| Siyasət | 14.890 | Political news, government activities |
| Mədəniyyət | 12.657 | Cultural events, artistic endeavors |
| Qan yaddaşi | 11.125 | Historical events and discussions, commemorations |
| Dünya | 4.293 | International news, global events |
| İdman | 2.580 | Sports-related news, athletic events |
| Cümhuriyyət | 0.486 | Content related to the centennial celebration of the republic |
| Müsahibə | 0.022 | Featuring discussions with individuals from various fields |
| Baş xəbərlər | 0.010 | Most important and significant news stories |
| Şuşa ili | 0.004 | Articles related to the year of Shusha |

## Language model

The OSCAR dataset (*Ortiz Suárez, Romary & Sagot, 2020*), which is a comprehensive collection of texts from multiple languages, served as the pre-training dataset. Utilizing RoBERTa as the architecture of choice, we pre-trained it using the OSCAR dataset on the MaskedLM task. To establish a baseline for comparison, we fine-tuned our language model using the original Azerbaijani news dataset. This baseline model provides a foundation for evaluating the effectiveness of our augmentation strategies. The imbalance of the class distribution and sentiment representation within the dataset posed challenges for the performance of the baseline model. The complete pipeline of the proposed method is shown in Fig. 3. Moreover, the trained model was published for public usage (*Hub, 2023*).

## Augmentation

The original text was translated into English to apply the EDA augmentation technique. Following the translation, the dataset underwent further expansion through the application of "EDA: Easy Data Augmentation Techniques." To create a balanced dataset, we applied augmentation only to classes with a small number of samples. We applied Synonym Replacement, Random Insertion, Random Swap, Random Deletion with same alpha parameter equal to 0.1 and generated 9 augmented sentences each original sentence. The differences between the augmented and original datasets are presented in Table 2. We implemented three distinct translation approaches to augment our dataset, as shown in Fig. 4. Class distribution comparison shown in Fig. 5.

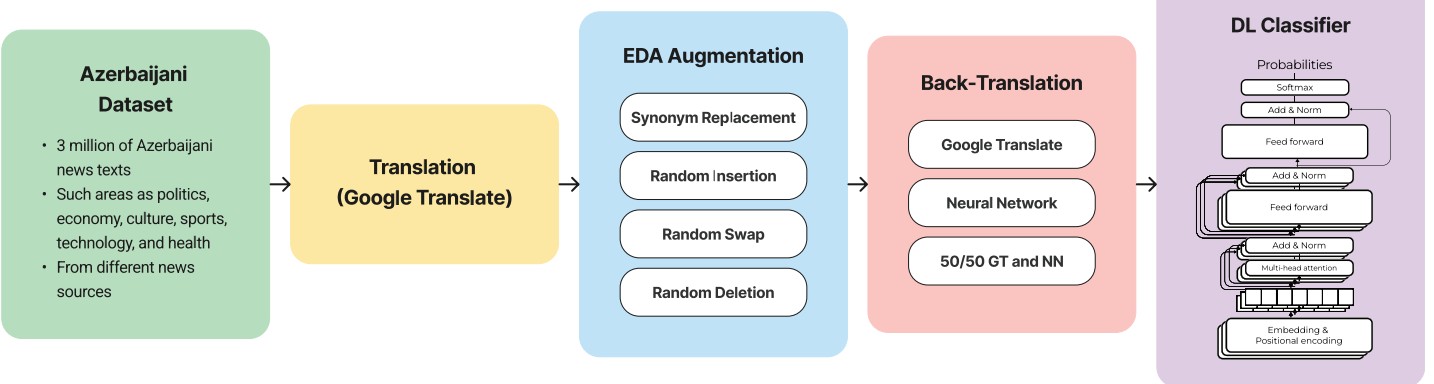

**Figure 3 Pipeline of the method.**

**Table 2 Difference in the number of sentences in each augmented class before and after augmentation.**

| Class name | Before | After | % increase |
|---|---|---|---|
| İdman | 27,721 | 304,930 | 1000,00% |
| Cümhuriyyət | 5,220 | 57,420 | 1000,00% |
| Müsahibə | 233 | 2,565 | 1000,86% |
| Baş xəbərlə | 105 | 1,155 | 1000,00% |
| Şuşa ili | 40 | 440 | 1000,00% |
| Qan yaddaşi | 119,544 | 13,12,690 | 998,08% |
| Dünya | 46,127 | 507,365 | 999,93% |

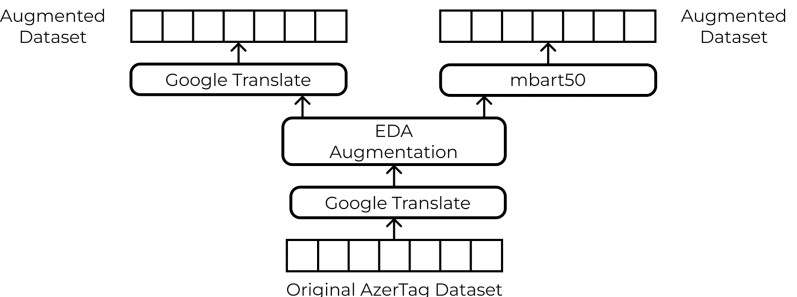

**Figure 4 Augmented dataset creation pipeline.**

### Translation with Google Translate API

Using this approach, we leveraged Google Translate for the translation. We translated sentences from Azerbaijani into English, applied EDA, and then translated them back into Azerbaijani. Augmented sentences were incorporated into the training data to enhance the diversity of the dataset and reduce class imbalance.

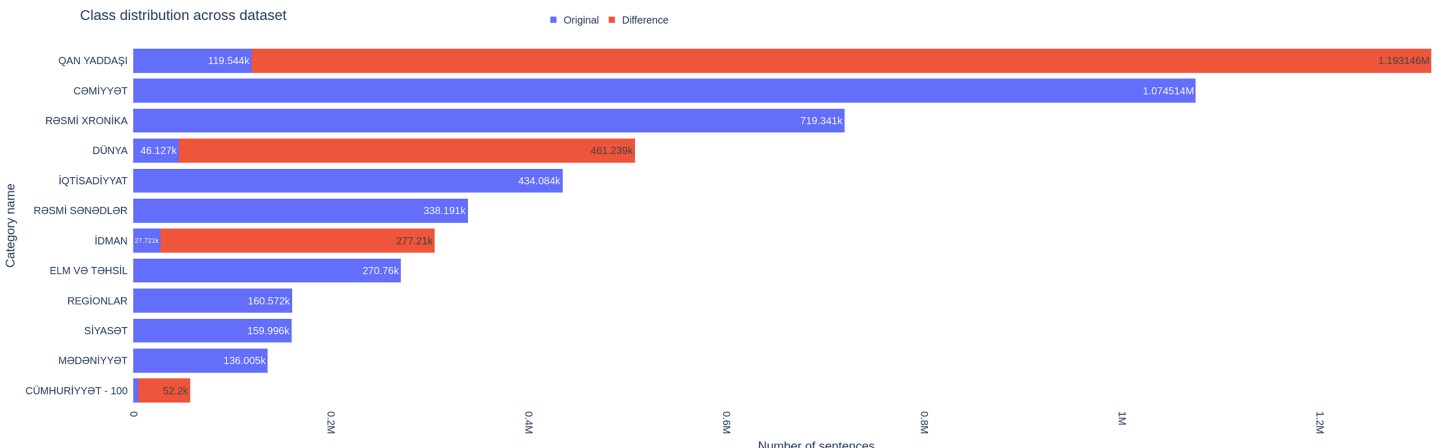

**Figure 5 Changes in distribution of sentences across different classes between original and augmented dataset.**

### Translation with mbart50

To translate sentences, we used a pre-trained mbart50 (*Tang, Tang & Yuan, 2020*) model, which follows the scheme of BART (*Lewis et al., 2020*). It is based on a standard sequence-to-sequence Transformer architecture with 12 encoder and 12 decoder layers. This approach aim to introduce synthetic sentences while maintaining the linguistic characteristics of Azerbaijani news texts.

### 50/50 translation with Google Translate API and mbart50

This hybrid approach combines Google Translate with mbart50. In this approach, we used both translations in equal proportions, thereby combining the benefits of both methods and further enhancing richness of the dataset.

## Text classification

To evaluate the efficacy of the augmentation techniques, RoBERTA with a classification layer was fine-tuned and evaluated for each dataset. The performance of the model was evaluated by comparing it with a baseline model trained on the original dataset. The evaluation metrics included the accuracy, precision, recall, and F1-score.

## Experimental setup

### System configuration

Our computational framework was equipped with the following hardware specifications:

- CPU: 32 cores, 64 threads
- Memory: 128 GB RAM
- GPU: 2 × NVIDIA GeForce RTX 2080 Ti
- Programming Language: Python 3.6.7
- Deep Learning Framework: PyTorch version 1.10.1

- NLP Library: Huggingface Transformers version 4.10.0

### Language model pre-training

The RoBERTa language model was pre-trained using the Oscar *corpus* (version 2301), which involved a training process spanning three epochs with a total of 27,198 iterations. The batch size was set to 32 per GPU, without gradient accumulation. The pre-training utilized dual NVIDIA GeForce RTX 2080 Ti GPUs, optimizing training efficiency. The default learning rate and optimizer settings of the RoBERTa model were employed. Model checkpoints were saved after the completion of each epoch. The standard RoBERTa tokenizer with its default vocabulary size was used.

### Fine-tuning for text classification

For the text classification task, the RoBERTa model was fine-tuned to handle 15 distinct labels. The learning rate was set at 2e-5, and both the training and evaluation batch sizes were fixed at 16 per device. Fine-tuning was executed over two epochs with a weight decay set at 0.01 to prevent overfitting. The evaluation strategy involved assessing the model's performance at the end of each epoch, and model checkpoints were saved correspondingly.

## RESULTS AND DISCUSSION

Our primary goal was to assess the impact of different text augmentation strategies, particularly those involving back-translation, on the performance of a multi-class classification model using the RoBERTa architecture. We employed a dataset consisting of 15 news topic categories, with varying degrees of augmentation to address the class imbalance issues.

### Baseline performance

We established the baseline performance by training the RoBERTa model on the original dataset without any augmentation. The baseline accuracy was 83%, reflecting the model's ability to distinguish between different news topics, as shown in Table 3.

### Augmentation strategies and performance

We evaluated three distinct augmentation strategies, each involving back-translation and original dataset. A comparison of different versions of dataset is presented in Table 4 and Fig. 6 correspondingly.

### Back-translation using google translate (GT)

Employing Google Translate, we back-translated the augmented English dataset into Azerbaijani. This approach resulted in an accuracy of 87%, further highlighting the contribution of augmentation to the model performance. Similar to NN-based back-translation, the precision, recall, and F1-scores exhibited noticeable enhancements across multiple categories.

### Back-translation using neural network (NN)

In this approach, we utilized a neural network-based translation process to convert the augmented English dataset back to Azerbaijani. The augmented dataset demonstrated an

**Table 3 Baseline performance.**

| Class | Precision | Recall | F1-score | Support |
|---|---|---|---|---|
| Cəmiyyət | 0.82 | 0.88 | 0.85 | 214,903 |
| Rəsmi xronika | 0.83 | 0.88 | 0.85 | 143,868 |
| İqtisadiyyat | 0.86 | 0.74 | 0.80 | 5,544 |
| RAsmi sənədlər | 0.39 | 0.26 | 0.31 | 1,044 |
| Elm və Təhsil | 1.00 | 0.02 | 0.04 | 47 |
| Regionlar | 0.00 | 0.00 | 0.00 | 21 |
| Siyasət | 0.00 | 0.00 | 0.00 | 8 |
| Mədəniyyət | 0.85 | 0.81 | 0.83 | 86,817 |
| Qan yaddaşi | 0.94 | 0.93 | 0.93 | 67,638 |
| Dünya | 0.80 | 0.71 | 0.75 | 54,152 |
| İdman | 0.88 | 0.80 | 0.83 | 32,115 |
| Cümhuriyyət | 0.69 | 0.53 | 0.60 | 31,999 |
| Müsahibə | 0.77 | 0.74 | 0.75 | 27,201 |
| Baş xəbərlər | 0.72 | 0.79 | 0.75 | 23,909 |
| Şuşa ili | 0.68 | 0.50 | 0.58 | 9,225 |
| Accuracy | | | 0.83 | 698,491 |
| Macro avg | 0.68 | 0.57 | 0.70 | 698,491 |
| Weighted avg | 0.83 | 0.83 | 0.83 | 698,491 |

**Table 4 Comparison between results of baseline and models fine-tuned on augmented datasets.**

| Type class | Baseline F1-score | Google translate | | Neural network | | 50/50 | |
|---|---|---|---|---|---|---|---|
| | | F1-score | Delta | F1-score | Delta | F1-score | Delta |
| İdman | 0.80 | 0.94 | **+0.14** | 0.89 | **+0.09** | 0.90 | **+0.10** |
| Cümhuriyyət | 0.31 | 0.79 | **+0.48** | 0.72 | **+0.41** | 0.68 | **+0.37** |
| Müsahibə | 0.04 | 0.75 | **+0.71** | 0.49 | **+0.45** | 0.50 | **+0.46** |
| Baş xəbərlər | 0.00 | 0.30 | **+0.30** | 0.04 | **+0.04** | 0.11 | **+0.11** |
| Şuşa ili | 0.00 | 0.58 | **+0.58** | 0.20 | **+0.20** | 0.22 | **+0.22** |
| Qan yaddaşi | 0.75 | 0.95 | **+0.20** | 0.93 | **+0.18** | 0.94 | **+0.19** |
| Dünya | 0.58 | 0.92 | **+0.34** | 0.83 | **+0.25** | 0.89 | **+0.31** |
| **Accuracy** | 0.83 | 0.87 | **+0.04** | 0.86 | **+0.03** | 0.86 | **+0.03** |
| **Macro Avg** | 0.59 | 0.77 | **+0.18** | 0.70 | **+0.11** | 0.70 | **+0.11** |
| **Weighted Avg** | 0.83 | 0.87 | **+0.04** | 0.86 | **+0.03** | 0.86 | **+0.03** |

**Note:**
Only those classes that have been augmented are shown. The increase compared to the baseline is highlighted in bold.

improved accuracy of 86%, indicating the effectiveness of this technique in enhancing the model's predictive capabilities. The precision, recall, and F1-score metrics exhibited improvements across various news categories.

It should be noted, that this method is less accurate than the method that employs the Google Translate API, but as the Google Translate API is a paid service with rate limits, this method is significantly more cost- and time-effective.

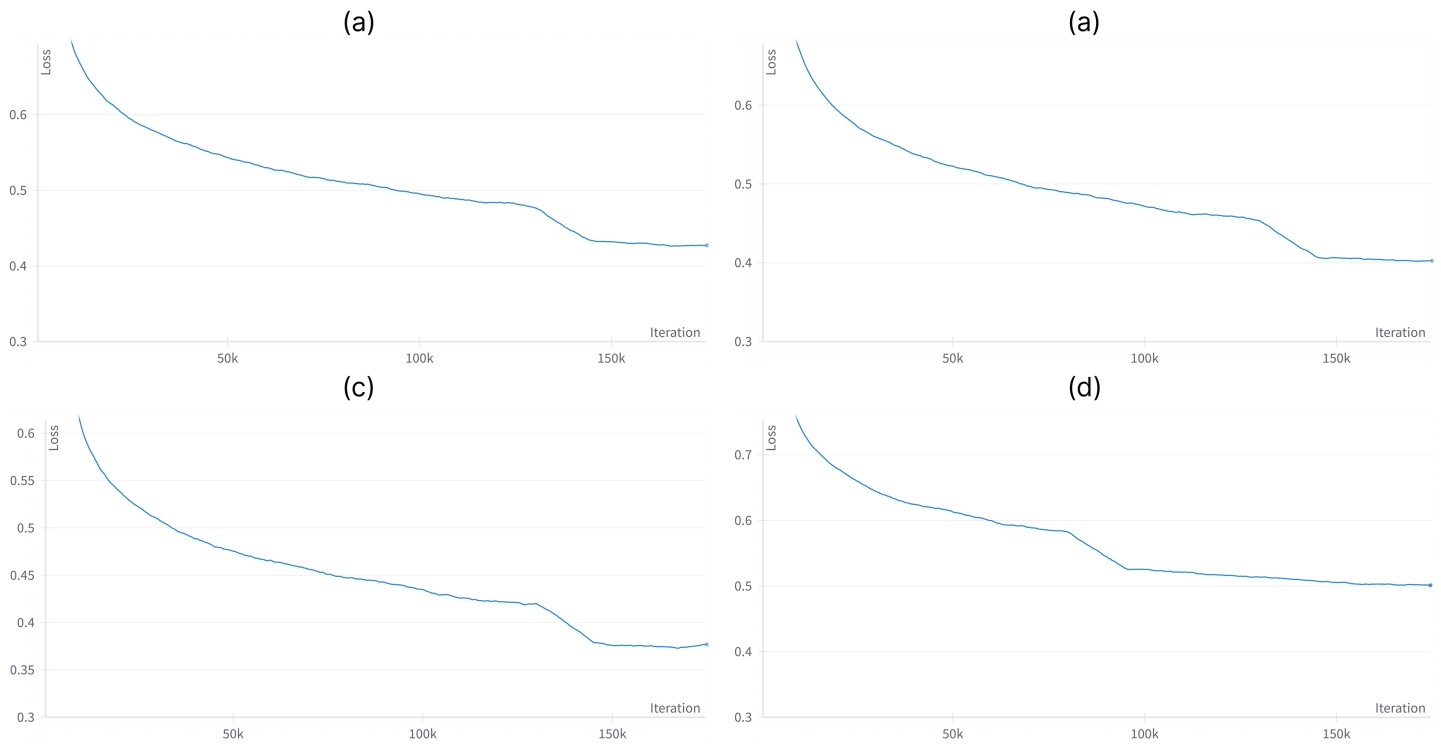

**Figure 6  Performance on news text classification tasks for augmented and original datasets.** (A) Augmented dataset with mBart-50 translation. (B) Augmented dataset with 50%/50% translation. (C) Augmented dataset with Google Translate API translation. (D) Original non-augmented dataset.

### Combined augmentation (50/50 NN and GT)

In this strategy, we utilized a 50/50 combination of neural networks and Google Translate-based back-translation. This balanced approach yielded an accuracy of 86%, validating the effectiveness of multiple back-translation techniques. The precision, recall, and F1-scores improved compared with the baseline, but the results were not substantial enough to justify the use of that approach.

### Analysis of findings

The comparison findings presented in Table 4 indicate that the most effective approach involves incorporating translation through the utilization of Google Translate API, with an accuracy rate of 87%. We can see that across all methodologies, there was a substantial increase in the F1-score for the augmented classes. Notably, in four classes of six augmented, the growth in the F1-score demonstrated an average increase of at least one-third or more. Furthermore, it is important to note that for "Baş xəbərlər" and "Şuşa ili" within the baseline approach, the F1-score was recorded as 0, indicating that the model exhibited utter failure to accurately identify items belonging to these specific classes. In terms of accuracy, the approach employing the Google Translate API exhibited the largest improvement (+0.04), whereas the other two methods demonstrated a gain of 0.03.

**Table 5 The main findings and contributions of the study.**

| Type | Statement | Deliverables |
|---|---|---|
| Contribution | Introduced the first publicly available transformer-based language model for Azerbaijani language. | *Hub (2023)* |
| Contribution | Proposed the labelled dataset of Azerbaijani News publicly available | *AzerTac (2023)* |
| Contribution | First, to the best of our knowledge, demonstrated the effective application of various text augmentation techniques for enhancement news classification task in Azerbaijani language | Section Introduction |
| Finding | Evidenced the effectiveness of Neural Machine Translation (mBart-50) with an F1-score of 0.86, highlighting its competitive quality against Google Translation Service while offering faster processing and cost-efficiency. | Table 4 |
| Finding | Observed that the Google Translate strategy achieved a notable result with a score of 0.87, indicating a 0.04 improvement. | Table 4 |
| Finding | The mixed approach resulted in an accuracy of 0.86, closely aligned with the mBart-50 translation strategy, suggesting limited benefits relative to the resources and time expended. | Table 4 |

Nevertheless, it is important to acknowledge that the remaining two approaches exhibited an improvement in the F1-score.

### Impact on imbalanced classes

Augmentation is particularly beneficial for small classes and effectively alleviates class imbalance issues. The precision, recall, and F1-scores for such classes showed notable enhancements, highlighting the potential of augmentation to addressing challenges related to underrepresented categories. A comparison of those metrics and loss between different versions of datasets is presented in Table 4 and Fig. 6, respectively.

In this study, we applied augmentation techniques exclusively to a subset of classes. Consequently, our evaluation focused on the impact of these techniques in mitigating class imbalance to a certain extent. In future studies, it would be beneficial to examine the outcomes of completely eliminating class imbalance and extending the augmentation approach to the entire dataset, rather than solely targeting imbalanced classes.

### Comparative analysis

The augmentation strategy involving a balanced combination of neural network and Google Translate-based back-translation yielded the most favorable results. This suggests that leveraging diverse translation sources can lead to more comprehensive augmentation.

## CONCLUSION

The findings of our research indicate that augmentation techniques, specifically those involving back-translation, can substantially enhance the performance of a RoBERTa-based multi-class classification model for Azerbaijani news text. The augmentation strategies demonstrated their effectiveness in improving the accuracy, precision, recall, and F1-scores, with the Google Translate strategy achieving the most promising results, with a score of 0.87, representing a 0.04 gain. On the other hand, the neural network strategy achieved a score of 0.86, which, although slightly lower, offered wider accessibility and faster execution. The mixed approach showed an accuracy of 0.86, which is similar to that of the Neural Network strategy. Moreover, augmentation is particularly valuable for addressing class imbalance concerns and enhancing the model's generalization

capabilities. These outcomes highlight the potential of text augmentation techniques to improve the accuracy and reliability of text classification tasks for low-resource languages, such as Azerbaijani.

Additionally, this study highlights that neural network-based translation shows comparable quality to Google Translate, with the added advantages of faster processing time due to unrestricted request frequency and cost-free usage. Consequently, it substantiates the viability of employing neural network-based translations from low-resource languages which opens the possibility of various NLP studies in such languages, particularly in Azerbaijani. The labeled text classification dataset and the pre-trained RoBERTa model for the Azerbaijani language were made publicly available following the completion of our research. The main findings and contributions are summarized in Table 5.

## Role of back-translation

Although we cannot check whether it would be better or worse if we could augment the data without back-translation, we can still conclude that even with double translation, augmentation shows an increase in the classification quality.

## Impact on low-resource languages

Our study focuses on Azerbaijani, a low-resource language in terms of NLP resources, underscoring the importance of augmentation techniques in such contexts. Augmentation, combined with back-translation, offers a viable approach for overcoming limitations related to data scarcity and linguistic complexities. These results suggest that augmentation can enable the creation of more robust and effective classification models for languages with limited training data and resources.

## Methodological limitations and advantages

This study, centered on enhancing the effectiveness and generalization of news text classification models through text augmentation techniques, entails specific methodological considerations. Despite the innovative approach, it is crucial to articulate both the limitations and strengths of the methods employed to provide a comprehensive understanding of the study.

### Limitations

**Language model and data representativeness** The use of the OSCAR dataset for pre-training the RoBERTa model, while advantageous for its extensive multilingual content, presents limitations in capturing the unique linguistic nuances of Azerbaijani. This could impact the model's adaptability and precision for Azerbaijani news text classification.

**Translation and augmentation techniques** Relying on translation services such as Google Translate and Facebook mBart50 model, along with Easy Data Augmentation (EDA) techniques, introduces potential risks of contextual misrepresentation and semantic loss. These methods, although effective in augmenting data volume, might not accurately preserve the linguistic integrity of the original Azerbaijani texts.

**Class imbalance and sentiment representation** The initial imbalance in class distribution and sentiment representation within the dataset challenges the baseline model's performance, possibly leading to biases or skewed classifications in augmented datasets.

### Advantages

**Enhanced dataset diversity** The augmentation process significantly expanded the dataset, potentially improving the model's ability to handle a diverse range of text classifications and reducing the impact of initial class imbalances.

**Innovative use of technologies** The application of advanced NLP technologies like RoBERTa, Google Translate API, and mbart50 for Azerbaijani, a language with limited computational resources, demonstrates the study's contribution to the field of language processing.

**Practical implications** The publication of the trained model for public use offers practical advantages for further research and applications in Azerbaijani text classification, setting a precedent for future studies in underrepresented languages.

## Future research directions

The success of augmentation techniques has encouraged further research in several respects. We intend to extend the examination of our proposed methodology to additional low-resource languages in future investigations. Investigating the optimal ratio of neural network-based and external translation services for back-translation could provide deeper insights into achieving the most effective augmentation. Additionally, exploring domain-specific augmentations tailored to news content could lead to even more specialized models.

### Funding

This work was supported by the Ministry of Education and Sciences of the Republic of Kazakhstan under the following grants: #AP14871214 "Development of machine learning methods to increase the coherence of text in summaries produced by the Extractive Summarization Methods" and #AP09260670 "Development of methods and algorithms for augmenting input data for modifying vector word embeddings". The funders had no role in study design, data collection and analysis, decision to publish, or preparation of the manuscript.

### Grant Disclosures

The following grant information was disclosed by the authors:
Ministry of Education and Sciences of the Republic of Kazakhstan: #AP14871214 and #AP09260670.

### Competing Interests

The authors declare that they have no competing interests.

## Author Contributions

- Atabay Ziyaden conceived and designed the experiments, performed the experiments, performed the computation work, prepared figures and/or tables, authored or reviewed drafts of the article, and approved the final draft.
- Amir Yelenov conceived and designed the experiments, performed the experiments, authored or reviewed drafts of the article, and approved the final draft.
- Fuad Hajiyev analyzed the data, authored or reviewed drafts of the article, and approved the final draft.
- Samir Rustamov analyzed the data, prepared figures and/or tables, authored or reviewed drafts of the article, and approved the final draft.
- Alexandr Pak conceived and designed the experiments, authored or reviewed drafts of the article, and approved the final draft.

## Data Availability

The raw data are available at Zenodo: Rustamov, S. (2024). AZERNEWSV1: AZERBAIJANI NEWS CLASSIFICATION DATASET [Data set]. Zenodo. https://doi.org/10.5281/zenodo.10638520.

The main jupyter notebooks are available at GitHub and Zenodo:

-https://GitHub.com/iamdenay/azerbaijani_augmentation/releases/tag/paper

- iamdenay. (2024). iamdenay/azerbaijani_augmentation: Zenado archive (paper). Zenodo. https://doi.org/10.5281/zenodo.10638680.

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
