# Peer review of "Text data augmentation and pre-trained Language Model for enhancing text classification of low-resource languages"

_PeerJ Computer Science, doi:10.7717/peerj-cs.1974_

## Round 0.1 · original submission · Major Revisions

Dear authors
Reviewers have now commented on your article and suggest major revisions. When submitting the revised version of your article, it will be better to address the following concerns and suggestions clearly.

1. The contributions of this paper are not clear. I have read the abstract and the introduction, but I have not understood the main contribution of this work. I suggest that you put considerable effort into improving the main contribution of this research. The paragraphs on related works just show the text of related works without any effect, discussion or overview to give a summary for this text.

2. Please, provide a paragraph with three to five clear positive impacts of the proposed method.

3. You have just described the related work that the researchers have done, but you have not evaluated the advantages and disadvantages of the related work. With regard to the introduction, it is not specified what makes this article different from the other studies available in the literature. The gap in the existing literature by arguing what is missing or inadequate in the existing solutions and therefore your study is necessary is not identified. This needs to be briefly stated and then further developed with in-depth analysis and justification of citations.

4. Important data are missing in the experimental results section so that the experiments can be reproduced, and even so that conclusions can be drawn from the reported results. For example, basic questions as the number of runs that have been carried out for each experiment are not mentioned in this section or in the rest of the paper. All of the parameters proposed and used methods are not clearly presented.

5. Add further details on how simulations were conducted. Similarly, system and resource characteristics could be added to Tables for clarity. The paper lacks the running environment, including software and hardware. The analysis and configurations of experiments should be presented in detail for reproducibility. It is convenient for other researchers to redo your experiments and this makes your work easy acceptance. A table with parameter setting for experimental results and analysis should be included in order to clearly describe them.

6. Clarifying the study’s limitations allows the readers to better understand under which conditions the results should be interpreted. A clear description of limitations of a study also shows that the researcher has a holistic understanding of his/her study. However, the authors fail to demonstrate this in their paper. The authors should clarify the pros and cons of the methods. What are the limitation(s) methodology(ies) adopted in this work? Please indicate practical advantages, and discuss research limitations.

Reviewer 1 ·

Basic reporting

The presented paper is devoted to the highly relevant task of developing native models for languages with insufficient resources. The proposed methodology is clear and efficient. The literature and references provided in this paper are interconnected, forming a comprehensive and well-supported foundation for the research. However, it can be improved if the authors include more closely related papers in the area of pretrained language models. The authors note, 'Based on our comprehensive analysis of the literature, it appears that no data augmentation approach has been used for the Azerbaijani language before. Our goal was to develop robust models through augmentation and fine-tuning strategies, capable of enhancing text classification tasks in Azerbaijani.' It would be interesting to investigate whether their approach is novel or if it has been applied to other languages for similar text classification tasks, and if so, to provide references to those sources.

Experimental design

Experimental design is well defined and highly compelling.

Validity of the findings

The conclusions are clearly articulated, closely aligned with the initial research question, and primarily focused on substantiating the obtained results.

Reviewer 2 ·

Basic reporting

No comment

Experimental design

No comment

Validity of the findings

No comment

Additional comments

The papers describes the results of experiments on data augmentation for low resource languages, specifically Azerbaijani and for the task of news article classification. Google Translate and mBART50 are used for a three-part pipeline for data augmentation. The work also includes creating some labelled data for news classification.

The authors should point out, for the record and to put the work in larger context, the state of the art (in terms of published or publicly reported results) for data augmentation for new classification with citation. I understand that this is first such work for Azerbaijani, but it will help readers in relating this paper with other similar work, especially for other low resource languages.

---

## Round 0.2 · Minor Revisions

Dear authors,

Thank you for your submission. Although you addressed many of the concerns and questions of the reviewers, one reviewer has some suggestions for the quality of the paper. We encourage you to address the concerns and criticisms of this reviewer and resubmit your article once you have updated it accordingly.

Best wishes,

Reviewer 1 ·

Basic reporting

Unfortunately, I could not find the answers to my review. So it seems that the authors respond to the comments of other reviewers.

"The presented paper is devoted to the highly relevant task of developing native models for languages with insufficient resources. The proposed methodology is clear and efficient. The literature and references provided in this paper are interconnected, forming a comprehensive and well-supported foundation for the research. However, it can be improved if the authors include more closely related papers in the area of pretrained language models. The authors note, 'Based on our comprehensive analysis of the literature, it appears that no data augmentation approach has been used for the Azerbaijani language before. Our goal was to develop robust models through augmentation and fine-tuning strategies, capable of enhancing text classification tasks in Azerbaijani.' It would be interesting to investigate whether their approach is novel or if it has been applied to other languages for similar text classification tasks, and if so, to provide references to those sources."

Experimental design

There are no comments for experimental design.

Validity of the findings

Findings are consistent with the goal.

Additional comments

Please provide answer for my main comment. It would be interesting to investigate whether their approach is novel or if it has been applied to other languages for similar text classification tasks, and if so, to provide references to those sources."

---

## Round 0.3 · accepted · Accept

Dear authors,

Thank you for the revision and for clearly addressing all the reviewers' comments. Your paper is now acceptable for publication in light of this last revision.

Best wishes,

Reviewer 1 ·

Basic reporting

Dear authors since you have informed that your approach quite novel I have no any additional comments.

Experimental design

Dear authors since you have informed that your approach quite novel I have no any additional comments.

Validity of the findings

Dear authors since you have informed that your approach quite novel I have no any additional comments.

Additional comments

Dear authors since you have informed that your approach quite novel I have no any additional comments.